# Historical and Scientific Evidence for the Origin and Cultural Importance to Australia’s First-Nations Peoples of the Laboratory Accession of *Nicotiana benthamiana*, a Model for Plant Virology

**DOI:** 10.3390/v14040771

**Published:** 2022-04-08

**Authors:** Steve Wylie, Hua Li

**Affiliations:** Plant Biotechnology Research Group (Virology), Western Australian State Agricultural Biotechnology Centre, Murdoch University, 90 South Street, Murdoch 6150, Australia; perthmuzi@yahoo.com

**Keywords:** model plant, Australian indigenous plants, Pituri, Australian Aborigines, Warlpiri, Solanaceae, nicotine, bioprospecting

## Abstract

*Nicotiana benthamiana* is an indigenous plant species distributed across northern Australia. The laboratory accession (LAB) of *N. benthamiana* has become widely adopted as a model host for plant viruses, and it is distinct from other accessions morphologically, physiologically, and by having an attenuation-of-function mutation in the RNA-dependent RNA polymerase 1 (*NbRdr1*) gene, referred to as *NbRdr1m*. Recent historical evidence suggested LAB was derived from a 1936 collection by John Cleland at The Granites of the Northern Territory, although no scientific evidence was provided. We provide scientific evidence and further historical evidence supporting the origin of LAB as The Granites. Analysis of a herbarium specimen of *N. benthamiana* collected by Cleland in 1936 revealed that The Granites population contains plants heterozygous for the *NbRdr1* locus, having both the functional *NbRdr1* and the mutant *NbRdr1m* alleles. *N. benthamiana* was an important cultural asset actively utilised as the narcotic *Pituri* (chewing tobacco) by the Warlpiri Aboriginal people at the site, who prevented women of child-bearing age from consuming it. We propose that Aboriginal people selected some of the unique traits of LAB that have subsequently facilitated its adoption as a model plant, such as lack of seed dormancy, fast maturity, low nornicotine content, and gracility.

## 1. Introduction

### 1.1. Genus Nicotiana 

*Nicotiana* L. is a large genus within the nightshade family Solanaceae, consisting of at least 76 species classified into 13 sections [1]. The genus was named after Jean Nicot de Villemain, a French ambassador to Portugal who introduced tobacco to France in 1560 [2]. The evolutionary centre of origin of the genus is the Americas, and its most (in)famous species, *Nicotiana tabacum*, tobacco, is indigenous to tropical and subtropical parts of the Americas, including the Caribbean, where it and other *Nicotiana* species have been used for thousands of years as narcotics and for medicinal and spiritual reasons [3,4,5]. *Nicotiana* species have probably been used for the same reasons by the Aboriginal peoples in Australia [6,7,8,9] for some or all of the 65,000 years humans have occupied the continent, although evidence of its use is not well preserved in the archaeological record. The use of Australian species of *Nicotiana* in dried or fresh leaf form as chewing tobacco by Australian Aboriginal peoples continues to the present day [10]. 

Nicotine is one of a number of pyridine alkaloids, including anatabine and anabasine, produced in *Nicotiana* roots and transported through the xylem to the leaves, and nornicotine, which is produced in leaves from nicotine. Alkaloid synthesis is elicited quickly via a systemic transcriptional response to insect herbivory [11]. Nicotine and other alkaloids serve as defensive compounds against insects through stimulation of acetylcholine receptors. In humans, nicotine stimulates dopamine release in the brain and is highly addictive for some people [12].

Species within the *Nicotiana* section *Suaveolentes* are indigenous to the Australian mainland (about 32 described species), some Pacific Islands (3 species), and a few mountains in Namibia (1 species) [13,14,15]. How colonisation by *Nicotiana* of Australia, Pacific Islands, and southwestern Africa was achieved is unknown. All Australian *Nicotiana* species are natural allopolyploids that resulted from a single hybridisation event of two diploid (*n* = 12) parental species to create *n* = 24. Since then, the ancestral allotetraploid species evolved into the range of species present today in Australia and elsewhere, adapting to diverse environments of the arid central and northern deserts, northern monsoonal regions, the wet tropics, and the temperate zones of the southern mainland continent, but not Tasmania [8]. Diploidisation has occurred where fusion or translocation of chromosomes occurred, resulting in a range from *n* = 15 to *n* = 24 [16]. *N. benthamiana* is *n* = 19. 

Very little is known about the reproductive biology of Australian *Nicotiana* species. Many appear to be mainly self-pollinated, but the light-coloured tubular flowers open in the evening and some emit fragrance at night, typical of moth-pollinated *Nicotiana* species in the Americas [17], although moth-borne pollination of Australian *Nicotiana* species has not been confirmed.

### 1.2. Nicotiana and Aboriginal Culture in Australia

*N. benthamiana* and other *Nicotiana* species remain of cultural significance to first-nations peoples of Australia. The dried leaves and stems of several *Nicotiana* species and *Duboisia hopwoodii* (family Solanaceae) are used to produce a form of chewing tobacco. Mixed with alkaline wood ash, this is referred to as *Pituri*, *Boodjerrie*, *Ingulba*, and *Mingkulpa*, among other names, which was an important trading commodity [6,7,8,18,19]. The first historical reference to this product was from William Wills’ diary, dated 7 May 1861, where he wrote ‘*They also gave us some stuff they call Bedgery or Pedgery. It has a highly intoxicating effect, when chewed even in small quantities. It appears to be the dried stems and leaves of some shrub*.’ [20]. Australian tobacco was not commonly smoked as it was in the Americas, although the smoke was used as an anaesthetic to numb the senses during surgical operations [21]. Dried *Nicotiana* and *Duboisia* leaves were also used to stupefy emus by adding them to small waterholes [22,23]. 

### 1.3. Nicotiana benthamiana and Science

*N. benthamiana* is a model plant used in many laboratories around the world. It’s susceptibility to viruses and fungi was first noted by tobacco breeders in the USA [24,25]. Because it is relatively easy to manipulate in vitro, *N. benthamiana* also provides a vehicle for transient and stable expression of transgenes and recombinant proteins, and for experiments using CRISPR-Cas9 and other gene ‘editing’ systems [26,27,28,29]. The Pubmed database lists over 5300 publications that refer to *N. benthamiana*. The release of partial genomes and transcriptomes of the laboratory accession of *N. benthamiana* has enhanced the scientific importance of this species [30,31,32,33]. 

With a few exceptions, descriptions in the scientific literature of experiments utilising this species refer to it simply as *Nicotiana benthamiana*; there is no accompanying accession code or variety name as is usually provided for descriptions of other model plants, such as *Arabidopsis thaliana*. An exception was Van Dijk and colleagues [34] who described two named accessions provided by the Tobacco Research Laboratory, North Carolina, as *N. benthamiana*–9 and -9A, and one from the Research Institute for Plant Protection, Wageningen, named *N. benthamiana*–IPO. Although no details of origins or distinct morphological characteristics of the three accessions were provided, all the plants responded in an identical manner to challenge by eight viruses, suggesting they were closely related or identical.

Goodin and colleagues [28] investigated the genetic diversity of five laboratory accessions of *N. benthamiana* sourced from laboratories in the USA, UK, and Spain using amplified fragment length polymorphisms. These five accessions were assigned the names Research Accession (RA) 1 to 5, and these, along with five others from the Kentucky Tobacco Research and Development Centre had similarity coefficients averaging 0.924, indicating they were all derived from the same recent common ancestor. The authors extrapolated these results to suggest that all laboratory accessions of *N. benthamiana* in use around the world were probably derived from a single source, although this has not been confirmed experimentally with plants from more sources nor with higher-resolution genotyping techniques. The accession used in our laboratory was derived from seed imported from the UK in the 1980s. Goodin’s Research Accession 4 (RA-4) was also collected from a UK laboratory, and so we arbitrarily adopted the name RA-4 for our accession to distinguish it from other *N. benthamiana* accessions in our possession [35,36]. On the other hand, Bally et al. (2015) [37] referred to their *N. benthamiana* line as ‘the laboratory accession’, or simply ‘LAB’, and for simplicity, we adopt this name here. 

In our experience, LAB differs from wild accessions of *N. benthamiana* in gross morphology (Figure 1) and in responses to infection by a tobamovirus. Generally, LAB plants are of shorter stature, they are more gracile (they have thinner and more flexible stems and petioles, smaller softer leaves, smaller flowers), flower colour is white (we observed a range of flower colours from cream to yellow to pink and purple), the leaves are of a lighter shade of green with entire margins whereas some wild accessions are of darker green undulate margins, and there are smaller softer trichomes on adaxial surfaces of leaves. Infection by yellow tailflower mild mottle virus induced a systemic hypersensitive response in LAB plants, whereas other accessions displayed varying degrees of symptom severity, but rarely death [35,36]. Similarly, Bally and colleagues (2015) [37] reported that LAB plants died when infected with the tobamovirus tobacco mosaic virus, while four wild-collected accessions did not.

The most significant characterised genetic difference between LAB and other accessions of *N. benthamiana* is the RNA-dependent RNA polymerase 1 (*NbRdr1* or *NbRDR1*) gene. *Rdr1* genes in many plant species are up-regulated in response to salicylic acid, jasmonic acid, and other phytohormones, and play roles in transcriptional regulation and control of viruses [38]. *NbRdr1* of LAB contains an insertion mutation of 72 nucleotides, as is referred to as *NbRdr1m* to distinguish it from the wild-type *NbRdr1*. The mutation introduces translation stop codons leading to truncation of the protein product [39]. It is not known whether the truncated RDR1 retains RNA polymerase activity. Studies with other plant species showed RDR1 is a component of an RNA–interference (RNAi) pathway that is active against virus invasion [40]. Bally and colleagues [37] proposed that progenitors of LAB maintained the mutant allele *NbRdr1m* because it conferred faster reproduction times and larger seed, an adaptation to the aridity and uncertain soil moisture conditions of northern Australian deserts, selected at the cost of loss of RNAi-based virus defence. However, *Rdr1* in the related species *Solanum tuberosum* (potato) was not reliably expressed in response to infection by three viruses, including a tobamovirus, suggesting *Rdr1* may not have a major role in virus defence, or that there is redundancy between it and other *Rdr* genes [38]. 

*NbRdr1m* was not found in other *N. benthamiana* populations tested from the Western Australian mainland, Barrow Island, or the Northern Territory [35,36], with the apparent exception of one plant from an undisclosed location in South Australia [37].

### 1.4. Geographical Origin of LAB 

Bally and colleagues [41] referred to correspondence between Professor John Cleland of the University of Adelaide in South Australia and Professor Thomas Goodspeed at the University of California (UC) at Berkley revealing the likely geographical origin of LAB as The Granites, Northern Territory. The correspondence referred to was located in the archives of UC Berkeley and is an undated hand-written note, apparently from Goodspeed’s assistant (signature unclear) informing Goodspeed that a letter received from Cleland (not shown) the previous day contained a packet of *N. benthamiana* seed ‘*from the locality he mentions*.’ The locality is not stated in the note. 

The aim of this paper is to examine further historical as well as scientific evidence for the geographical origin of *N. benthamiana* LAB. Correct identification of the *N. benthamiana* population from which LAB was sourced is vitally important because it will enable a study of the parental population to determine how wild plants carrying *NbRdr1m* respond to viral and other challenges in the wild, and it may provide clues as to why this apparently rare recessive allele is maintained in a natural system.

## 2. Materials and Methods

### 2.1. Origin of Historical Sample

Leaf material and seed was obtained from one of the *N. benthamiana* plants John Cleland collected at The Granites on the 25 August 1936, hereafter referred to as *N. benthamiana* Granites. This specimen was lodged in the Australian National Herbarium in Canberra by Cleland, and its origin is recorded as *The Granites, 400 miles NW of Alice Springs, coordinates* −*20.5667, 103.35, Tanami Region of the Northern Territory*. The catalogue number is CANB112241.1. A sample of this plant consisting of dry leaf fragments and about 200 seeds was kindly provided to the authors in 2016 by Dr Brendan Lepschi, Curator, Australian National Herbarium, 80 years after Cleland collected it. Another four *N. benthamiana* samples collected by Cleland from The Granites site on the same trip in 1936 and one sample described as being collected 20 miles south of The Granites are lodged with the State Herbarium of South Australia under accession codes AD95711022, AD95711023, AD97219292, AD97615134, AD97615135. Although a number of requests were made of the State Herbarium of South Australia for samples of these five *N. benthamiana* plants, our requests were not fulfilled and no explanation was given. 

### 2.2. Seed Germination

Approximately 100 seeds from *N. benthamiana* Granites were surface sterilised by soaking in 2% sodium hypochlorite solution for 2 min, drained, and 70% ethanol added for 1 min before thoroughly rinsing the seeds several times in sterile distilled water. One hundred fresh seeds of our *N. benthamiana* laboratory accession RA-4 (LAB) were treated in the same way. The surface-sterilised seeds were spread on sterile 0.5X Murashige and Skoog agar medium [42] in Petri dishes. The cultures were maintained under natural light at room temperature (16–20 °C). 

A further 100 seeds of *N. benthamiana* Granites and fresh RA-4 seeds were sprinkled directly onto damp bark/sand potting mix without the surface treatment described above, and a clear polythene bag placed over the pots to maintain humidity during germination. The pots were incubated in a climate-controlled greenhouse where the temperature was maintained at 22 °C during the day and night without artificial lighting. 

### 2.3. DNA Extraction

DNA was extracted from 20 mg of dried *N. benthamiana* Granites leaf tissue using two methods. The DNeasy Plant Mini kit (Qiagen) was used according to the manufacturer’s protocol for dried plant materials, and a CTAB-based method [43] was used. The DNeasy method was also used to extract total DNA from 100 mg fresh leaf material of *N. benthamiana* RA-4 (LAB), *N. benthamiana* MtA-6 collected from the Mount Augustus region in Western Australia (−24.306344, 116.914120) [35], and an F1 hybrid between *N. benthamiana* RA-4 (LAB) and MtA-6 plants. DNA was quantified on a nanodrop spectrophotometer (Thermo Fisher, Waltham, MA, USA).

### 2.4. Crossing NbRdr1m and NbRdr1 N. benthamiana Accessions

F1 hybrids of RA-4 (LAB) (*NbRdr1m*/*NbRdr1m*) and MtA-6 (*NbRdr1/NbRdr1*) plants were produced by emasculating RA-4 flowers before anther dehiscence and applying pollen collected from recently dehisced MtA-6 anthers to RA-4 stigmas. Each pollinated flower was protected by covering with a small paper bag. Seed was collected from individual capsules when they opened. 

### 2.5. PCR Amplification

The insertion-mutation site *NbRdr1m* was amplified using primers RP1 (5′-TTTCTTGCATTTTCATCGAGC-3′) and RP2 (5′-AATGAAGTTATCTTTGAGAAATAG-3′). Between 10 and 50 ng of total DNA was used as a template for amplification by PCR. Amplification conditions were an initial denaturation step of 95 °C for 2 min, followed by 30 cycles of 95 °C for 10 s, 50 °C for 20 s, 72 °C for 30 s. PCR amplification resulted in either a 484 bp product from the *NbRdr1m* allele, or a 413 bp product from the *NbRdr1* allele. PCR products were separated on 1% agarose gels stained with Sybr Green (SigmaAldrich). Before sequencing, individual bands were excised from agarose gels and purified using a QIAquick Gel Extraction kit (Qiagen) spin column following the manufacturer’s protocol. Purified PCR products were cloned in pGemT-easy vector (Promega), and both DNA strands were sequenced from the vector by Sanger sequencing using T7 and SP6 primers. 

### 2.6. Historical Evidence

Historical records of *N. benthamiana* samples lodged in state or national herbaria were obtained from The Australian Virtual Herbarium (https://avh.chah.org.au, accessed on 19 July 2020).

## 3. Results

### 3.1. Seed Germination

Four weeks after sowing, no *N. benthamiana* Granites seeds had germinated in vitro or in pots. Over 80 of the RA-4 (LAB) seed had germinated both in vitro and in pots. It is assumed that all seed from Cleland’s plant had lost viability in the 80 years since its collection. 

### 3.2. PCR Amplification and Sequencing

Amplification of *N. benthamiana* Granites DNA extracted using the DNeasy kit resulted in low yields of the amplified product. The amplification of DNA extracted by the CTAB method provided a higher yield. Surprisingly, in both cases, the pattern of amplification was identical to that of the F1 hybrid between *N. benthamiana* accessions RA-4 (LAB) and MtA-6, showing that *N. benthamiana* Granites was heterozygous for *NbRdr1* and *NbRdr1m* alleles (Figure 2). The bands were excised individually from the gel, cloned and sequenced. Sequencing confirmed the presence of the 72 nt insertion mutation in *N. benthamiana* Granites as described by Yang et al. [39]. 

### 3.3. Historical Evidence 

After his death, Sir John Burton Cleland (1878–1971) (Figure 3) was described as ‘the last of the gentleman naturalists’ [44]. John Cleland’s title was Professor of Pathology at the University of Adelaide, South Australia. He trained as a medical doctor but developed an extraordinary range of expertise, including microbiology, parasitology, epidemiology, pathology, zoology, mycology, botany, and anthropology [45]. He was appointed to an advisory committee to the Aborigines Department in 1933, Deputy Chairman of the South Australian Aborigines Protection Board from 1938, and a member of the Aborigines Advisory Board from 1962 to 1965. He undertook field work to remote communities and wrote extensively on Aboriginal health and how they interacted with the environment [23], including use of native plants and fungi for food and medicine [18,46,47,48,49,50]. He also collected many specimens of the indigenous fauna and flora of Australia, which remain lodged in museums and herbaria today. For *Nicotiana* species alone, he lodged 241 specimens over his lifetime, his first being *N. maritima* in 1898 when he was 20 years old, and his last, also *N. maritima*, in September 1968 when he was 90 years old and blind (AVH, 2018). 

#### 3.3.1. Cleland and Johnson’s Trip to The Granites in 1936

In August 1936, John Cleland, Thomas Harvey Johnson (1881–1955), Professor of Zoology and Professor of Botany at the University of Adelaide [51], and others embarked on a 4000 km expedition from Adelaide in South Australia to The Granites in the Northern Territory.

The Granites is a series of low granite hills in the Tanami Desert, Northern Territory, approximately 540 km northwest of Alice Springs and 500 km southeast of Halls Creek. It has an arid subtropical climate with distinct wet and dry seasons. The wet season is between November and March (approximately 350 mm precipitation annually), and temperatures range from a mean summer maximum of 39 °C (102 °F) to mean winter maximums of 24–29 °C (75–84 °F) [52].

The expedition to The Granites in 1936 was one of several annual expeditions Cleland embarked on. They were all organised by the Board for Anthropological Research of the University of Adelaide and the South Australian Museum and were largely funded by the Rockefeller Foundation through the Australian National Research Council. The primary reason for the 1936 expedition was to study medical and cultural aspects of the Warlpiri Aboriginal people (Cleland’s spelling was *Wilpirri*) who were still living a largely traditional life at The Granites (*Boorkitji*) and to record their use of and names of plants at the site. Cleland also collected plants while travelling there, including other *Nicotiana* species, and he recorded sightings of birds [53]. On 13 of August 1936, he collected his first *N. benthamiana* specimen, lodged at the State Herbarium of South Australia in Adelaide (catalogue code AD95711023), from a site 20 km to the south of The Granites (−20.86, 130.35) before reaching The Granites via the Tanami Road. They camped near the Warlpiri camp, probably at *Yartalu Yartalu* (−20.57213, 130.35370) located next to a hill of granite boulders. There were about 50 Walpiri people living there, whom Cleland described as ‘*nearly all nude and living their natural nomadic life*’ [54]. Camping nearby and also studying Warlpiri culture and painting the flora of the region was Ms Olive Muriel Pink (1884–1975), anthropologist and botanical artist [55]. Cleland acknowledged her help, writing that she ‘*engaged in social anthropology in the locality and … facilitated our enquiries*’. Cleland reported finding *N. benthamiana* plants growing ‘*luxuriantly amongst the rocks on granite knolls*’. He described the Warlpiri people’s use of this plant: ‘*The leaves were first moistened by being chewed for a short time, and then the mass is rolled in ashes obtained by burning twigs of Hakea (probably also of Acacia)… The quid is chewed for its narcotic properties and eventually passed to other natives*. *The roll is carried behind the ear when not in use.*’ He noted that ‘*leaves were eaten by young and old men and the old women, but not by young girls*. *We observed one old woman fill her mouth with the fresh leaves and swallow the bolus*’. He recorded the children’s name for *N. benthamiana* plants as *Tukkamulla* while adults called it *Tangungnu* [56]. The *Hakea* species used for ash was *H. cunninghamii, Wakilbirri*, which was also used to make boomerangs by the Warlpiri. Notably, Cleland did not witness the leaves being baked or dried for production of *Pituri* as recorded in other places [6,7].

On 24 August, he collected five more *N. benthamiana* plants from among the granite boulders near their camp site, which he pressed and dried and later lodged in the Australian National Herbarium in Canberra (catalogue code CANB112241.1) and the State Herbarium of South Australia in Adelaide (catalogue codes AD95711022, AD97219292, AD97615134, AD97615135). The plants he chose were flowering and had mature seed pods. This second collection site was close to the present-day Newmont Tanami Operations gold mine and Granites airport.

#### 3.3.2. Is The Granites the Geographical Origin of LAB?

It is unlikely that Cleland had seen *N. benthamiana* before he visited The Granites site. Of the 241 *Nicotiana* specimens Cleland lodged in herbariums over the course of his life, only 7 were of *N. benthamiana*: the 6 specimens collected from The Granites area (above) in 1936 and 1 specimen (AD966041616) collected from the Gibson Desert, Western Australia, in June 1958 (AVH, 2018).

A letter in the archives of the University of California at Berkeley library indicates that in 1939 Cleland sent a packet of *N. benthamiana* seed from a Granites plant to Professor Thomas Harper Goodspeed (1887–1966) at UC Berkeley [41,57]. At the time, Goodspeed was the Director of the University of California Botanical Garden located in Strawberry Canyon on the UC Berkeley campus and was considered the world authority on genus *Nicotiana*. He maintained a large collection of *Nicotiana* species, growing them in the botanical garden. Cleland had previously sent seed of other Australian *Nicotiana* species to Goodspeed (see below).

Did Goodspeed already have *N. benthamiana* plants in his possession prior to receiving the seed sent by Cleland in 1939? It seems not. In 1935, Ms Helen Mar Wheeler, a graduate student under the supervision of Goodspeed, formally described 14 Australian species of *Nicotiana*, including naming a new species, *N. goodspeedii*, in honour of her PhD supervisor [58]. The 14 species were variously described from herbarium specimens, from botanical drawings, and from live specimens already growing in the UC Berkeley Botanical Gardens. She attributed three species to Cleland: *N. excelsior* and *N. gossei* from specimens collected by Cleland in August 1932, and *N. maritima* from a plant collected by Cleland in September 1932. Of interest here is Wheeler’s description of *N. benthamiana*, which she described based on a herbarium specimen from the Victoria National Herbarium collected in 1883, and from Domin’s 1929 drawing of the type specimen collected on the third voyage of HMS Beagle (1837–1843) by William Boyne [59]. Wheeler’s formal description of *N. benthamiana* was not from a live specimen. She would have used a live specimen if one was available to her, so it is almost certain that *N. benthamiana* was not present in Goodspeed’s collection in 1935. Thus, The Granites-derived *N. benthamiana* seed that Cleland sent to Goodspeed in 1939 was the first of that species to be included in Goodspeed’s collection. Cleland sent no further *N. benthamiana* seed to Goodspeed, and as far as is known, neither did anyone else.

Goodspeed subsequently distributed the seed either directly from the packet Cleland sent him, or more likely, from fresh seed collected from plants grown in the UC Berkeley Botanical Gardens to the Division of Tobacco Investigations, Beltsville, Maryland. Six years after Goodspeed received the seed from Cleland, *N. benthamiana* was first reported as an experimental host of the ‘the mosaic virus’ of tobacco (tobacco mosaic virus) [25] and of the pathogenic fungus of tobacco *Peronospora tabacina* [24]. The source of *Nicotiana* seed used in the fungal experiment was acknowledged as ‘*Largely through the courtesy of the University of California Botanical Garden*’. The molecular and historical evidence strongly supports The Granites as the geographical origin of LAB.

## 4. Discussion

Our findings support the hypothesis that LAB originated at The Granites [41], but many questions remain about this important plant. LAB differs from wild-collected *N. benthamiana* accessions in that it contains an insertion mutation in its *Rdr1* gene, named *NbRdr1m*. Professor Goodspeed at UC-Berkeley experimented with X-rays and radium to induce random mutations in the genomes of several species of *Nicotiana* [60], and we considered the possibility that *NbRdr1m* was generated artificially in Goodspeed’s laboratory. However, this hypothesis has now been rejected with confirmation that *NbRdr1m* existed naturally in The Granites population in 1936.

The herbarium specimen we tested (CANB112241.1) was heterozygous for the recessive *NbRdr1m* allele, whereas LAB is homozygous for this allele. We do not know which of the six specimens that Cleland collected was sent to Goodspeed, but given the apparent ubiquity of the homozygous recessive *NbRdr1m* genotype in laboratories today, we consider it likely that by chance Cleland sent Goodspeed seed from a homozygous *NbRdr1m* plant, and not of heterozygous CANB112241.1. An analysis of Cleland’s other five plants held by State Herbarium of South Australia would reveal whether homozygous *NbRdr1m* plants exist among them. Still, it is possible that Cleland sent Goodspeed a heterozygous plant from which homozygous recessive (LAB-like) genotypes were subsequently selected. 

Mendelian genetics predicts segregation ratios of 1:2:1 for single-copy genes upon self-fertilisation of the heterozygote. Thus, three *Rdr1* genotypes should exist at The Granites: *NbRdr1m*/*NbRdr1m* resembling LAB, *NbRdr1m*/*NbRdr1* resembling CANB112241.1, and *NbRdr1*/*NbRdr1* resembling other known *N. benthamiana* populations [35]. The relative fitness of the three genotypes under natural selection is unknown. Why would the apparently rare *NbRdr1m* allele be maintained in the region of The Granites but not in other areas? Bally and colleagues hypothesised that *NbRdr1m* disabled virus defence in favour of faster reproductive cycles, but others [36] showed LAB plants to be no more susceptible to virus infection and accumulation than wild-type *NbRdr1* plants, placing this hypothesis in doubt.

Bally and colleagues placed the mutation event of *NbRdr1* to *NbRdr1m* at 710–880 thousand years BP [37], long before humans occupied the continent 65,000 years BP [9]. If this were the case, it seems distribution of the allele either has not spread beyond the region it first occurred in or has shrunk to this region from a broader distribution over this vast time period. Our investigation of an accession that closely resembled LAB from Joe Creek located to the north of The Granites, and one from Mount Tietkens to the southeast revealed *NbRdr1m* was not present in either [35]. Thus, *NbRdr1m* could be localised in the vicinity, although more detailed testing of nearby populations is needed to confirm this. 

Could 65,000 years of human occupation of northern Australia have played a role in selection and maintenance of the unusual features of LAB, including the recessive *NbRdr1m* allele, to the present day? In 1960, Nancy Burbidge in describing *N. benthamiana* accessions from across its natural range wrote: ‘*a considerable range of variation is at present accepted under this name…*’, suggesting that *N. benthamiana* may, in fact, comprise more than one species/subspecies, all with 19 pairs of chromosomes [61]. Indeed, LAB plants differ in several visible ways from plants of other *N. benthamiana* accessions, as described above. Notably, LAB seeds lack the dormancy of wild plants; they germinate almost immediately after being shed, whereas seeds of wild plants remain dormant for weeks to months after shedding unless treated with gibberellic acid [36]. Seed dormancy is an important survival trait in wild plants but generally undesirable in crops and in laboratory plants. Although one could argue that loss of seed dormancy and other apparent ‘domestication traits’ [62] apparent in LAB were selected for by scientists in laboratories after 1936, we consider this scenario unlikely. The LAB progenitor seed received at UC Berkeley was likely from a single plant, and its progeny has been self-pollinated ever since. Thus, LAB is highly inbred with little to no heterozygosity [28] and therefore lacks the genetic breadth needed for selection of multiple apparent ‘domestication’ traits described above. An investigation of the wild The Granites population should confirm this.

We hypothesise that at least some traits distinctive to LAB were consciously selected by the Warlpiri people and their ancestors who used this plant as a narcotic over vast time periods. Nicotine is extremely addictive and users typically desire a constant source of the alkaloid. Plants with non-dormant seed would germinate almost immediately after capsule dehiscence if it fell to moist soil sheltered between granite boulders, perhaps enabling several generations of the fast-maturing LAB-like *N. benthamiana* plants per year and thereby providing a source of nicotine over a long period. It is not hard to imagine that gracile plants with relatively soft leaves (Figure 1c) lacking the thick leaves and prickly trichomes of ‘robust’ genotypes (Figure 1a) are far more comfortable to chew fresh. 

It is more difficult to imagine how humans could have unconsciously selected plants harbouring *NbRdr1m*. Bally and colleagues [37] proposed that natural selection maintained *NbRdr1m* by trading virus resistance for larger seeds and early maturity in uncertain soil moisture conditions. There are several problems with this hypothesis. *N. benthamiana* populations in the same climatic zone harbouring wild-type *NbRdr1* are far more prevalent than the rare LAB *NbRdr1m* genotype. There is no clear mechanistic link between *NbRdr1m* and early maturity and large seededness. LAB and wild-type accessions show no differences in tobamovirus susceptibility and accumulation [36].

If humans inadvertently selected *NbRdr1m* plants, which traits were being selected? *N. benthamiana* plants were a source of nicotine for the Warlpiri, so if human selection of *NbRdr1m* occurred, it was probably related to one or more of the ‘domestication’ traits (e.g. loss of seed dormancy, gracility) or it was directly related to its alkaloid content. Some other *Nicotiana* species and genotypes readily convert the alkaloid nicotine to nornicotine via N-demethylation in naturally senescing leaves and leaves that are dried and processed for consumption, whereas other genotypes of the same species are non-converters [63]. Nornicotine is undesirable in tobacco because of its off-taste and unpleasant aroma [64], but more importantly, because it is the precursor of N’-nitrosonornicotine (NNN), a potent carcinogen [65]. A study of alkaloid contents of Australian *Nicotiana* species revealed a wide range of nicotine and nornicotine concentrations in leaves, and wide differences between accessions of some species [66]. These researchers did not test LAB plants, instead using accession A109412 (*NbRdr1*) [35], which contained 2.3 mg/g nicotine and 0.25 mg/g nornicotine dry weight. An earlier study by Saitou and colleagues [67] did not name the *N. benthamiana* accession they analysed, but we can safely assume it was LAB because other accessions were not available at the time. They found (LAB) plants contained slightly more nicotine than did A109412 (2.9 mg/g vs. 2.3 mg/g), but LAB’s nornicotine content was far lower, only 10% (0.02 mg/g) of A109412 [67]. It is conceivable that Warlpiri removed plants tasting/smelling of high nornicotine content, perhaps aware of the (NNN) toxicity of such plants? They were certainly aware of the adverse maternal and perinatal outcomes of nicotine consumption by women of child-bearing age [68]. Cleland recorded, ‘*N. benthamiana leaves were eaten by young and old men and the old women, but not by young girls*’ (emphasis added). Drying and fermenting *N. tabacum* leaves promotes the conversion of nicotine to nornicotine, and Cleland did not observe leaves being dried or baked, as often occurred in the preparation of *Pituri* in other places in Australia [68]. An isoform in the cytochrome 450 gene family, *CYP82E4*, converts nicotine to nornicotine in *N. tabacum* [63]. Could there be a link between the *NbRdr1m* allele and the relatively low levels of nornicotine present in leaves of LAB? Plants of the North American species *N. attenuata* in which *Rdr1* (*NaRdr1*) was silenced had significantly lower levels of nicotine than isogenic wild-type plants [69], but nornicotine levels were not measured. Because nicotine levels in wild-type *N. benthamiana* accession A109412 (*NbRdr1*) were closely similar to those of LAB, it remains unclear whether the *NbRdr1m* gene influences nicotine and nornicotine levels in LAB. Measurement of levels of alkaloids in isogenic homozygous *NbRdr1m* plants, heterozygous *NbRdr1m/NbRdr1* plants, and homozygous *NbRdr1* plants may answer this question, and clarify if humans played a role in selection of *NbRdr1m*.

In 1936, there remained widespread ambivalence to the rights of Australia’s first-nations peoples by European colonists. Aborigines were not recognised as full citizens of Australia until 1967 [70]. When Cleland, Johnson, Pink, and others were studying the Warlpiri people at The Granites, it had been only eight years since a massacre of between 50 and 200 Warlpiri, Ammatyerr, and Kaytyete children, women, and men by white police and cattle farmers in and around Coniston Station, located to the south of The Granites, [71,72,73]. Mr Gwoja Tjungarrayi, a Warlpiri-Anmatyerre man, was one of the few survivors of the 1928 massacre that destroyed his family. The design on the current Australian AUD 2 coin was inspired by a drawing of Tjungarrayi (Figure 4). It was in this context that Cleland entered what had been Warlpiri land for thousands of years. With the colonial attitudes of the day, he felt perfectly entitled to remove specimens of *N. benthamiana* and other plants and take them back to Adelaide with him. It is likely that Cleland was unaware of customary Aboriginal practices of assigning harvesting rights over particular plants to certain individuals or groups, and of totemic, kinship, and spiritual associations that his actions may have impacted [74]. He certainly would not have considered that he was stealing intellectual property (IP). Today, traditional culture and knowledge are recognised and respected to a greater extent, although there are still significant gaps in intellectual property protection for Australian Aboriginal peoples [75]. Australia is a signatory to both the Convention on Biological Diversity (signed 1993) and the Nagoya Protocol (signed 2012). As a signatory, Australia has a responsibility to ensure fair and equitable sharing of benefits to indigenous communities arising from use of Australia’s biological resources. Once the germplasm has been removed from Australia, opportunities for negotiating benefit-sharing agreements decrease sharply. Nevertheless, removal from Australia of large collections of indigenous *Nicotiana* species for research continues to the present day without recourse to traditional owners or IP considerations [76]. Notably, *N. benthamiana* LAB is the subject of at least 80 patents in several countries, none of which acknowledge Aboriginal IP [77]. As climatic and other pressures impact agricultural systems everywhere, there is a focus on accessing Australian and other desert plants as sources of novel genes and microorganisms for crop improvement, for pharmaceuticals, and as food [78,79,80,81,82,83,84]. The accumulated dollar value to agriculture and industry of LAB alone is difficult to calculate, although it is many millions of dollars. Not surprisingly, none of this money has trickled back to the Warlpiri because LAB’s origin as The Granites has only recently been revealed. As wild relatives of some of the world’s most important food crops: potato, tomato, capsicum, eggplant, etc., the scientific value of *N. benthamiana* and other Australian solanaceous species is potentially immense.

## 5. Conclusions

We argue that human selection, probably over millennia, of a *N. benthamiana* population at The Granites for loss of seed dormancy and faster maturity have been critical in facilitating adoption of *N. benthamiana* LAB as a model plant for international science and industry. LAB continues to offer enormous benefit to mankind as a model plant, especially for virology, as a source of genes, as an easily transformed tool for understanding gene function, and as a factory for molecular ‘pharming’ of recombinant proteins [85,86,87]. Further study of the *N. benthamiana* population located in the region of The Granites and the surrounding land is of great scientific interest. This task remains challenging because of the remote location of the site, the uncertainty of rains and timing of germination of *N. benthamiana* seed, and the development of the site as a goldmine [88]. Of critical importance before such studies are undertaken is engagement with the traditional owners of the region.

## Figures and Tables

**Figure 1 viruses-14-00771-f001:**
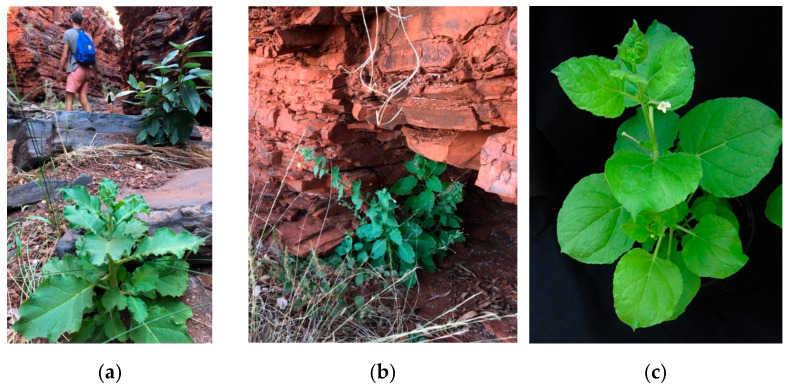
Wild plants of *N. benthamiana* growing in a shaded spot in damp gravel in a stream bed (**a**) and in the drip zone at the entrance to a cave (**b**) in northern Western Australia, both habitats typical of this species (July 2020). A plant of *N. benthamiana* LAB growing in a greenhouse (**c**). Photos S. Wylie.

**Figure 2 viruses-14-00771-f002:**
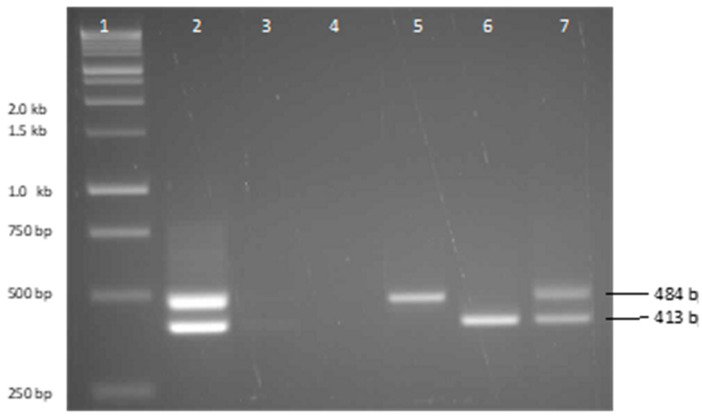
Amplification products of *Rdr*1 sequences from *N. benthamiana* plants. Lane 1, 1 kb DNA ladder (Promega). Lane 2, F1 progeny of a cross between RA-4 (LAB) and MtA-6 (*NbRdr1m/NbRdr1*) plants; Lane 3, *N. benthamiana* Granites herbarium sample amplified from DNA prepared using Qiagen Plant DNeasy kit method; Lane 4, negative control where water replaced genomic DNA; Lane 5, *N. benthamiana* RA-4 (LAB) (*NbRdr1m*); Lane 6, *N. benthamiana* MtA-6 (*NbRdr1*); Lane 7, *N. benthamiana* Granites herbarium sample prepared using a CTAB-based method.

**Figure 3 viruses-14-00771-f003:**
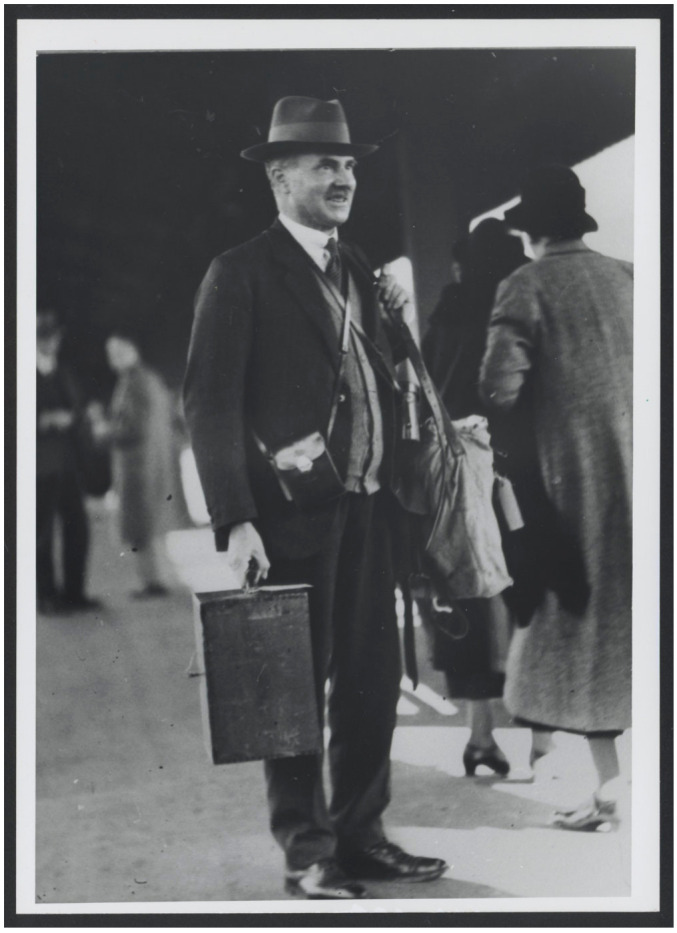
Professor John Cleland at age 56 at Adelaide railway station in 1934 anticipating an adventure. Photo State Library of South Australia, B33263.

**Figure 4 viruses-14-00771-f004:**
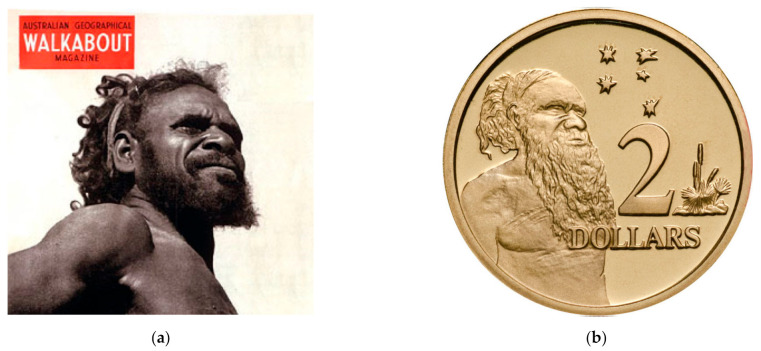
(**a**) A photograph of Warlpiri-Anmatyerre man Mr Gwoja Tjungarrayi (~1895–1965) taken in 1935 as it appeared on the cover of the September 1950 issue of Walkabout Magazine. Photographer Roy Dunstan. Courtesy Australian National Travel Association and Mitchell Library, State Library of New South Wales. (**b**) The Australian AUD 2 coin featuring an Aboriginal elder, inspired by an image of Mr Tjungarrayi.

## Data Availability

Not applicable.

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
