# Peer review of "Historical and Scientific Evidence for the Origin and Cultural Importance to Australia’s First-Nations Peoples of the Laboratory Accession of Nicotiana benthamiana, a Model for Plant Virology"

_viruses, 2022, doi:10.3390/v14040771_

Round 1

Reviewer 1 Report

The manuscript by Wylie and Li entitled“Historical and scientific evidence for the origin and cultural importance to Australia’s first-nations peoples of the laboratory accession of Nicotiana benthamiana, a model for plant virology”provides further historical as well as scientific evidence which strongly supports the geographical origin of laboratory accession (LAB) of Nicotiana benthamiana as a 1936 collection by John Cleland at The Granites of the Northern Territory (refferd to The Granites). Analysis of a herbarium specimen of N. benthamiana collected by Cleland in 1936 revealed that The Granites population contains plants heterozygous for the NbRdr1 (RNA-dependent RNA polymerase 1) locus, having both the functional NbRdr1 and the mutant NbRdr1m alleles. LAB plants differ from wild-collected N. benthamiana accessions in that it contains an insertion mutation in its Rdr1 gene (NbRdr1m). Correct identification of the geographical origin of N. benthamiana LAB might helps the study of the parental population to determine how wild plants carrying NbRdr1m respond to viral and other challenges in the wild, and provide clues as to why the recessive allele is maintained in a natural system.

I think it is worthy of publication in Viruses after minor revisions.

Specific comments:

  1. Line 216-231: The “Introduction shouldbe properly reduced to adapt the journal’s requirements.
  2. Line 26-152: The difference about the amplification yields betweenN. benthamianaGranites DNA extracted using the DNeasy kit and the CTAB method should be resulted from the dried plant materials of N. benthamiana Granites, which means the DNeasy kit is not applicable to the dried plant materials. My suggestion is that just extract all of the N. benthamiana plants DNA with the CTAB method, run PCR and change Figure 2.

Author Response

We thank reviewer 1 for taking the time to read the manuscript and make suggestions. We address each suggestion below. Our responses in bold"

The manuscript is attached with changes tracked as advised.

Line 216-231: The “Introduction shouldbe properly reduced to adapt the journal’s requirements.

We read the journal's requirement for the Introduction, appended below in italics. There is no length requirement, and we mention the aims and the main findings. We have not changed the text because it seems to meet the requirements of the journal.

Introduction: The introduction should briefly place the study in a broad context and highlight why it is important. It should define the purpose of the work and its significance, including specific hypotheses being tested. The current state of the research field should be reviewed carefully and key publications cited. Please highlight controversial and diverging hypotheses when necessary. Finally, briefly mention the main aim of the work and highlight the main conclusions. Keep the introduction comprehensible to scientists working outside the topic of the paper.

Line 26-152: The difference about the amplification yields betweenN. benthamianaGranites DNA extracted using the DNeasy kit and the CTAB method should be resulted from the dried plant materials of N. benthamiana Granites, which means the DNeasy kit is not applicable to the dried plant materials. My suggestion is that just extract all of the N. benthamiana plants DNA with the CTAB method, run PCR and change Figure 2.

Unfortunately, neither author has access to the laboratory to carry out the suggested work. This work was done in 2016 and our employment situations have changed. All the information is given in Fig 2, and we feel the difference in the two extraction systems may be of interest to readers who plan to extract DNA from decades-old stored leaf samples.

Reviewer 2 Report

This manuscript by Steve Wylie et al. reports the laboratory accession (LAB) of Nicotiana benthamiana having an attenuation-of-function mutation in the RNA-dependent RNA polymerase 1 (NbRdr1) gene (NbRdr1m) is a model for plant virology. By studying large number of historical materials and herbarium specimens, authors provide scientific and sufficient historical evidence supporting the geographical origin of LAB as a 1936 collection by John Cleland at The Granites of the Northern Territory. Furthermore, they propose that Aboriginal people may have unconsciously selected for NbRdr1m in the population, and consciously selected plants for low nornicotine content and lacking seed dormancy. Generally, compared with research articles related to gene function or interaction, this manuscript tells us the origin of N. benthamiana just like telling us an interesting story.

Some specific comments:

Introduction:

Line 32: Change “N. tabacum” to “Nicotiana tabacum L”.

Line 111: Is it possible to provide a morphological photo of LAB compared with the wild plant?

Line 115: please check “four wild-collected” or “our…”?

L214: Generally speaking, the seeds of tobacco are relatively easy to germinate, and N. benthamiana Granites seeds may germinate after vernalization induced by low temperature or interpret the reason of no germination of these seeds.

Lines 383-387 in the discussion section describe the morphological characteristics of LAB. I think it would be better to put the description in the introduction section.

Author Response

We thank reviewer 2 for their considered suggestions to improve the paper. Changes to the manuscript are tracked. Our responses to reviewer 2's suggestions are provided in bold (below):

Line 32: Change “N. tabacum” to “Nicotiana tabacum L”. Done as advised.

Line 111: Is it possible to provide a morphological photo of LAB compared with the wild plant? Yes, a photo of LAB has been provided in Figure 1, as requested, and another photo of wild plants to expand the range of wild habitats illustrated.

Line 115: please check “four wild-collected” or “our…”? four wild-collected is correct. It refers to Bally's wild-collected plants, not ours.

L214: Generally speaking, the seeds of tobacco are relatively easy to germinate, and N. benthamiana Granites seeds may germinate after vernalization induced by low temperature or interpret the reason of no germination of these seeds.

Herbarium specimens were maintained at low temperatures for decades (4oC), and so we did not consider lack of vernalisation an inhibitor to germination of these seeds. Seeds lose viability over time and we consider that the time factor is the most likely reason for lack of germination. We added the following sentence: It is assumed that all seed from Cleland’s plant had lost viability in the 80 years since its collection.

Lines 383-387 in the discussion section describe the morphological characteristics of LAB. I think it would be better to put the description in the introduction section. We agree, and have moved the descriptions of gross morphology to the introduction section of the paper.